# Association between Polymorphisms in the IL-1β, TNFRSF11B, CASP1, and IL-6 Genes and Orthodontic-Induced External Apical Root Resorption

**DOI:** 10.3390/jcm10184166

**Published:** 2021-09-15

**Authors:** Agata Ciurla, Crystal Marruganti, Tiziana Doldo, Jolanta Szymańska

**Affiliations:** 1Dentist’s Office ORTO-PUNKT, Mościckiego St. 72/1, 33-100 Tarnów, Poland; 2Unit of Periodontology, Endodontology, Restorative and Pediatric Dentistry, Department of Medical Biotechnologies, University of Siena, 53100 Siena, Italy; marruganti@gmail.com; 3Unit of Orthodontics, Department of Medical Biotechnologies, University of Siena, 53100 Siena, Italy; tiziana.doldo@unisi.it; 4Comprehensive Dentistry, Department of Comprehensive Paediatric and Adult Dentistry, Medical University of Lublin, 6 Chodźki St., 20-093 Lublin, Poland; szymanska.polska@gmail.com

**Keywords:** gene polymorphism, root resorption, orthodontic treatment

## Abstract

Orthodontic-induced external apical root resorption (EARR) is a severe condition affecting the roots of the teeth, whose genetic causes have been inconclusive to date. The aim of the present study was to assess the influence of selected single nucleotide polymorphisms (SNPs) IL-1β, TNFRSF11B, CASP1, and IL-6 genes on post-orthodontic EARR. A sample of 101 patients with clearly assessable orthopantomograms and lateral cephalometric radiographs taken before and at the end of the orthodontic treatment was used to evaluate the presence of EARR. The association between genetic polymorphisms and EARR was assessed with the Chi^2^ test. A binary logistic multi-level model was built to evaluate the ability of patient- and tooth-level variables to predict EARR occurrence. The overall prevalence of EARR resulted to be around 40%. Within the limitations of this study, a significant association was found between EARR presence and the SNP for the IL-1β gene but not for the TNFRSF11B, CASP1, and the IL-6 genes. The final multi-level model demonstrated that the SNP for the IL-1β gene increases the odds of developing EARR by around four times. Since there is currently no accurate method to determine which patients will develop EARR prior to orthodontic treatment, further studies are needed to investigate the predictive ability of further genetic variants on EARR development.

## 1. Introduction

External apical root resorption (EARR) is a multifactorial pathological process related to the irreversible resorption of the root structure. This is a frequent clinical complication of orthodontic treatment in response to a mechanical stimulus. However, its occurrence has been reported both with and without orthodontic treatment [1]. In 5% of orthodontic patients, advanced apical root resorption may develop, leading to significant root shortening of 5 mm or more [1]. Any unusual shape of the roots should be diagnosed during the initial examination. The risk of the excessive mobility of the involved teeth should be explained to the patient before the start of the treatment, even if EARR usually does not have a significant impact on teeth longevity [2].

EARR may develop in response to a variety of stimuli (e.g., mechanical, inflammatory, autoimmune, or infectious). The extent of root resorption depends on a combination of environmental and individual genetic factors. Genetic predisposition was described as a pivotal etiological factor for root resorption in several studies [3,4,5,6,7,8,9,10]. Such predisposition was reported for the first time in 1975 by Newman [11]. Next, Harris et al. described the involvement of genetic variation in EARR concurrent with orthodontic treatment-related factors through a heritability study [12]. Hartsfield et al. reported the association between the genetic variation, with 50–66% of the variation observed in EARR development after orthodontic treatment [9].

EARR concurrent with orthodontic treatment has been frequently associated with interleukin-1 genes, especially with IL-1β gene variability. Interleukin-1 is a potent pro-inflammatory cytokine, facilitating alveolar bone remodeling and, therefore, decreasing apical mechanical stress during orthodontic tooth movement [9,13]. The mechanical stimulus on the root is the main factor associated with post-orthodontic EARR, so that any unusual changes in IL-1 levels may affect EARR [9,14].

Another multifunctional cytokine with both inflammatory and anti-inflammatory effects is IL-6 (interleukin-6). In many studies investigating susceptibility to EARR, no correlation between IL-6 SNP and EARR was found; however, it was proven that the level of IL-6 in gingival crevicular fluid changes during tooth movement and that it is higher in tissues involved in root resorption [15,16,17]. Another genetic factor affecting EARR can be TNFRSF11B encoding osteoprotegerin. Hartsfield et al. reported a significant association between SNP rs2073618 of TNFRSF11B (OPG) gene and EARR concurrent with the orthodontic treatment [9]. CASP1 encoding caspase-1 is responsible for the activation of the two inflammatory agents, interleukin-1β (IL-1β) and interleukin-18 (IL-18), involved in the inflammatory process and root resorption; therefore, it was proposed as another candidate for susceptibility to EARR [18].

Despite the numerous studies, the relationship between CASP1, TNFRSF11B, IL-1β, and Il-6 SNP and EARR is still unclear and controversial; as such, it requires further research. The aim of the current study was to evaluate the contribution of genetic factors to the development of EARR in order to construct a multi-level model to predict EARR presence. The polymorphisms chosen for this study were rs1143634 from IL-1β, rs3102735 from TNFRSF11B gene, rs530537 from CASP1 gene, and rs1800796 from IL-6 gene; all of them have previously been associated with EARR and might be functionally relevant.

## 2. Materials and Methods

### 2.1. Study Design

The current study was reported according to the Strengthening the Reporting of Observational Studies in Epidemiology (STROBE) guidelines [19]. The protocol was approved by the Bioethical Committee of the Medical University of Lublin (KE-0254/335/2018).

### 2.2. Setting and Participants

Individuals were recruited among those coming to a private clinic (ORTO-PUNKT, Tarnòw, Poland) and undergoing fixed straight-wire orthodontic treatment between January 2019 and March 2021. The inclusion criteria were:–age between 18 and 30 years old;–teeth with a complete formation of the root;–patients of Polish Caucasian origin and not related to each other;–ability and willingness to give informed consent.

The exclusion criteria were:–presence of fractures or abrasions or incisal edges;–history of trauma in the selected teeth;–previous straight-wire orthodontic treatment;–presence of systemic diseases affecting the dental hard tissues.

### 2.3. Variables

Information regarding participants’ medical and dental history was registered. Moreover, orthopantomograms and lateral cephalometric radiographs were collected before and at the end of the orthodontic treatment.

#### 2.3.1. Radiographic Variables

Upper and lower central and lateral incisors, as well as upper first molars, were the teeth selected for examination. All measurements were carried out on digital orthopantomograms (OPG) taken before and at the end of treatment using a diagnostic software (Planmeca Romexis Viewer), as done in previous studies [3,4,7,12,20]. In order to assess the presence of EARR, loss in root length was assessed using the Linge method modified by Brezniak et al. [21,22,23]. The ratio (C1/C2) of crown length before (C1) and at the end of treatment (C2) was used, assuming that the length of the crown does not change throughout the orthodontic treatment [10]. The individual root-crown ratio (RCR) and the relative changes of RCR (rRCR) were calculated using a C1/C2 strengthening factor. rRCR ranges between 0 and 100%, where an rRCR of 100% means that no changes in root length occurred before (R1) and after (R2) the orthodontic treatment. In case any shortening of the root occurs, the rRCR decreases according to the equation rRCR = R1 − R2(C1 − C2) [24]. The variable rRCR was dichotomized as follows: (a) presence of EARR, if rRCR < 0.90; (b) absence of EARR, if 0.90 ≤ rRCR ≤ 1.00.

#### 2.3.2. Examiners Calibration

Measurements were taken by two calibrated orthodontists (C.A., S.J.). The reproducibility of measurements was statistically evaluated by comparing double measurements of 10 randomly selected X-rays with an interval between the respective measurements of 3 weeks. The mean error between measurements was calculated using the Dahlberg formula, as previously described [24]. The mean error in tooth length measurements was 0.33 mm. Moreover, the intraclass correlation coefficient (ICC) was calculated to evaluate examiners’ agreement.

#### 2.3.3. Sample Collection

Swabs from the oral cavity were collected with a disposable, sterile stub of swab stick with a transport medium. Each stub of swab stick containing the sample was placed in a sterile Eppendorf tube with 500 µL Lysis buffer and shaken on a vortex (2500 rpm for 1 min). After placing the samples in the refrigerator for about 30 min and stirring every 5 min, a stub of swab stick was removed, and the tubes were sealed. The Hirt two-days method was used to isolate DNA from the cell suspension. Samples with isolated DNA were centrifuged at 4 °C for 15 min at 13.2 thousand rotations per minute. Then, ethanol was completely removed from each sample, and 22 mL of ultra-pure nuclease-free water was added to each sample to dissolve the DNA. The samples were placed on ice. The concentration of isolated DNA was calculated on a precalibrated NanoDrop P2000C spectrophotometer. To determine DNA concentration, 2 µL of patient DNA to the measurement window was used to obtain a DNA concentration of 260 nm. The degree of purity of isolated nucleic acid is the factor (A260/A280)-the absorbance ratio at 260 nm to 280 nm. As the purity of the isolated DNA was in the range 1.8–2.05, the samples were taken for further analysis. Polymorphism of four genes was genotyped by PCR analysis, according to a previously published method [24].

### 2.4. Statistical Analysis

#### 2.4.1. Sample Size Calculation

The sample size was calculated through ad hoc statistical software (STATA BE version 17.1, StataCorp LP, College Station, TX, USA), setting the level of significance at 5%. Considering a 5% prevalence of EARR in the reference cohort [1] and a 15% in the study cohort, in order to achieve a power of 80%, the inclusion of 101 patients was planned.

#### 2.4.2. Descriptive and Inferential Statistics

Continuous variables were reported as Means ± Standard Deviation (SD) and 95% Confidence Interval (CI). Binary and categorical data were presented as proportions. Patient- and tooth-level variables, as well as the presence of genetic polymorphisms (IL-1β, TNFRSF11B, CASP1, IL-6), were compared according to the presence/absence of EARR using the Chi^2^ test and Mann–Whitney U test for categorical and continuous variables, respectively.

#### 2.4.3. Multi-Level Binary Logistic Regression

Given the presence of both tooth- and patient-level characteristics, the feasibility for the creation of a multi-level model was tested for the prediction of EARR (dependent variable), supposing that teeth (Level 1) were nested within patients (Level 2). First, in order to test this hypothesis, an intercept-only model was created. Afterward, a Likelihood Ratio (LR) test was performed to check whether the multi-level model represented a better fit than the single-level for EARR prediction. Evidence of significant clustering at the patient level was tested by calculating the intraclass correlation (ICC). Therefore, two levels were identified: Level 1 (tooth-level), with tooth type and arch as predictors, and Level 2 (patient-level) with age, gender, and genetic polymorphisms (IL-1β, TNFRSF11B, CASP1, IL-6) as predictors. The final model was chosen according to the lowest values of Akaike (AIC) and Bayesian (BIC) information criteria. The level of significance was set at alpha = 0.05 for all analyses.

## 3. Results

### 3.1. Participants Characteristics

A total of 101 patients were enrolled in the current study, and there were no drop-outs; therefore, all included patients were included in the analysis. The mean age was 21.32 ± 7.28 years, with a proportion of 76.24% females and 23.76% males (Table 1). After cephalometric analysis, 53 (52.47%) individuals were classified as class I, 41 (40.59%) as class II, and 7 (6.93%) as class III malocclusion. All included participants underwent a fixed orthodontic treatment (straight arch technique) lasting an average of 31.1 ± 6.4 months; in almost half of the patients (42.57%), a palatal expander (Hyrax) or a distalizer was used in the first phase of treatment. Moreover, class II camouflage with extractions of premolars was chosen in eight cases with class II malocclusion.

### 3.2. Outcome Variables

Examiners’ agreement resulted in an ICC of 0.90. In the current cohort, the majority of patients undergoing orthodontic treatment were not affected by severe root resorption. In fact, the prevalence of EARR below 90% of the root length concerned 9.21% of all the examined teeth. Serious EARR with the shortening of the root over 20% occurred in 2.28% teeth; the most commonly involved teeth were the right lateral upper (4.98% of all teeth) and the left central lower incisors (3.96% of all teeth). Advanced EARR of maxillary first molars was not observed. The 5.75% of examined teeth in the study group had advanced EARR with the reduction of the root length bigger than 20%, while in a proportion of 17.25% teeth, a shortening of 10–20% of their initial length occurred. Such results are reported in further detail in a previous publication [24].

The most frequent SNPs were the C/T polymorphism for both TNFRSF11B (23.76%) and CASP1 (20.79%). The C/G polymorphism for IL-6 was present in 10.89% of the cases, while the G/A polymorphism for IL-1β had a prevalence of around 9% (Table 1). No differences in age and gender were found between individuals with or without EARR (Table 2). EARR occurred more frequently in incisors (88.17%) than molars (11.83%) (*p* = 0.04), while no differences in EARR occurrence were found between upper and lower arch (*p* = 0.15). No statistically significant association was found between the presence of genetic polymorphisms for TNFRSF11B, CASP1, IL-6, and the occurrence of EARR (*p* > 0.05). On the other hand, the SNP for IL-1β was significantly associated with the presence of EARR (*p* = 0.03).

### 3.3. Multi-Level Binary Logistic Regression

The LR test demonstrated a significant improvement of the multi-level model compared to the single level (Table 3). Moreover, evidence of significant clustering of data at the patient level (grouping variable) was found (ICC = 0.48). Therefore, a multi-level binary logistic regression model was deemed appropriate. The final model obtained to predict the presence of EARR (dependent variable) included the following predictors: (a) tooth type (Level 1); (b) arch (Level 1); (c) IL-1β polymorphism (Level 2). Molars had significantly reduced odds of having EARR (*p* = 0.014), while the presence of the IL-1β polymorphism increased the odds of EARR by almost four times (*p* = 0.046) (Table 3).

## 4. Discussion

### 4.1. Summary of Findings

The current study demonstrated an overall prevalence of EARR of 40%, with no significant differences related to patients’ age or gender. None of the SNPs evaluated (TNFRSF11B, CASP1, IL-6) were significantly associated with EARR occurrence, except for the G/A polymorphism of IL-1β. The final multi-level analysis demonstrated that the odds of EARR occurrence were almost three times higher in incisors compared to molars and around four times higher whenever the G/A IL-1β polymorphism was present.

### 4.2. EARR Occurrence

Around 40% of the participants included in the current protocol presented signs of EARR, with around 90% of them being located in incisors rather than molars. In a previous study carried out in a cohort of 290 patients undergoing fixed orthodontic treatment, the prevalence of subjects presenting at least one tooth with a minimum of 1 mm resorption was 50.3% [25]; around 17.5% of subjects were found privy of any signs of resorption in any tooth [25]. The difference between the two studies may be due mainly to the type of radiograph used to assess EARR occurrence; in fact, while the previous study evaluated EARR through periapical radiographs, the current study used digital OPG. Consequently, the presence of EARR may have been underrated, especially when at its early stages. Other differences regarding patients’ age, evaluated teeth, and follow-up range may have played a role in the observed discrepancy in EARR prevalence. Furthermore, results from the current study are consistent with previous studies showing how EARR occurrence is higher in lateral incisors than central incisors or molars [25,26,27].

### 4.3. IL-1β, TNFRSF11B, CASP1, IL-6 Polymorphisms

The current study failed to identify a statistically significant association between EARR occurrence and SNPs for TNFRS11B, CASP1, and IL-6 polymorphisms. Many studies investigated the impact of genetic susceptibility to EARR and the release of inflammatory molecules involved in the inflammatory cascade.

IL-1β is a multipotential cytokine involved in the bone remodeling associated with orthodontic tooth movement. It was found that fluctuations in IL-1 concentration correlate with individual differences in tooth translation, which may contribute to EARR susceptibility. Such differences can be attributed in part to alleles of the polymorphic IL-1β gene since the IL-1β allele 2 at position +3954 is associated with a 4-fold increase in IL-1β production [4]. The studies by Al-Qawasmi et al. showed that the IL-1β polymorphism associated with EARR in patients undergoing orthodontic treatment is also correlated with the rate of IL-1 production in vitro. In particular, allele 1 of the IL-1β polymorphism at position +3954 is associated with relatively low IL-1β production [4]. Monocytes from people homozygous for the IL-1β +3954 allele 2 produce four times more IL-1β, and heterozygous cells produce about two times more IL-1β than cells from people homozygous for allele 1. The IL-2 allele found to be -1B +3954 is associated with adult periodontitis [28]. This is consistent with the observation that excessive IL-1 production activates the degradation of the extracellular matrix and bone in periodontal tissues.

Since IL-1β is a potent stimulus for bone resorption and recruitment of osteoclastic cells during orthodontic tooth movement, it has been hypothesized that low IL-1β production in the one allele may result in relatively less bone modeling (resorption) in the cortical bone. IL-1β inhibits the resorptive response to orthodontic pressure. Excessive root resorption associated with the IL-1β allele 1 may be due to a reduced rate of bone resorption at the interface of the periodontal ligaments [4].

Since Al-Qawasmi et al., for the first time, reported the genetic association of IL-1β polymorphisms with the clinical manifestation of EARR, this cytokine attracted the attention of many researchers [4]. The correlation between IL-1β +3953 polymorphism and EARR was confirmed in both an association study using family trios and a linkage study including 35 white American families. Patients with a homozygous combination of IL-1β +3953 allele C had a 5.6-fold increased risk of EARR compared to individuals with heterozygous genotypes [4]. Bastos Lages et al. examined 61 Brazilian patients treated with orthodontic appliances and showed that carriers of IL-1β +3953 C allele were at an increased risk of EARR conversely to the case-control study of Gülden et al., which showed no difference between the alleles of IL-1B [7,8]. Furthermore, contrary to the previously published results, Gülden et al. observed the allele 1 more often in the control group [8]. The Japanese data provided by Tomoyasu and the Czech data provided by Linhartova also questioned the association of IL-1β (+3953 C/T) with EARR [29,30]. In turn, Iglesias-Linares et al. demonstrated a positive correlation between the allele TT of IL-1RN single-nucleotide polymorphism (SNP) rs419598 and EARR [10].

*TNFRSF11B* (tumor necrosis factor receptor superfamily, member 11b) is a gene encoding osteoprotegerin (OPG), also known as a tumor necrosis factor receptor superfamily member 11B, which is secreted by osteoblast lineage cells of bone and plays a pivotal role in the regulation of bone metabolism [31]. OPG inhibits osteoclastogenesis and bone excessive bone resorption by binding to RANKL and preventing it from binding to RANK [32]. The OPG encoding gene (TNFRS11B) was also investigated in previous studies [33,34] due to its role in bone metabolism. In particular, a retrospective analysis carried out on 195 Portuguese individuals on OPG radiographs highlighted a lack of association between EARR and SNPs for the OPG-encoding TNFRS11B gene [21]; results were consistent with those obtained more recently by Linhartova et al. [33] and those reported in the current study.

CASP1 (Caspase 1) is a gene encoding caspase-1, which is an enzyme that proteolytically cleaves other proteins, such as the precursors of the inflammatory cytokines interleukin-1β and interleukin-18, into active mature peptides [34]. The CASP1 gene was previously investigated due to its role in the activation of the pro-inflammatory response and root resorption [18]. A previous study performed on pre- and post- radiographs of 460 individuals highlighted how EARR occurrence in maxillary incisors was significantly associated with the SNP for the purinergic-receptor-P2X, ligand-gated ion channel 7 (P2RX7) but not with polymorphisms related to CASP1 and IL-1β [35]. Results regarding the CASP1 gene, but not IL-1β, are consistent with those obtained in the current report. On the other hand, we demonstrated a significant association between EARR occurrence and SNP for IL-1β. Such inflammatory agent was often associated with inflammatory events in bone and connective tissue and is considered as a key element in the complex inflammatory pathway leading to root resorption [3,4]. Al-Qawasmi et al. suggested that IL-1β stimulates osteoclasts during orthodontic movement and that low IL-1β production may result in less absorption of dense bone at the border of the periodontal ligament [3,4]. Therefore, root resorption may result from necrosis of the periodontal ligament as an effect of prolonged stress concentrated in the root due to a slower rate of bone resorption. The relationship between EARR and IL-1β polymorphism, verified in the study by Lages et al., suggested the involvement of this cytokine in the etiopathology of EARR [7]. Patients with allele 1 (G/A) were more predisposed to EARR, and subjects homozygous for allele 2 (either G/G and A/A) were more protected against EARR. The association of IL-1βb with EARR occurrence in the study by Lages et al. was confirmed in the present research [7]. In fact, the multi-level model demonstrated how EARR presence was almost four times more likely in subjects with the heterozygous variant for IL-1β (G/A) compared to the homozygotic ones (either G/G or A/A).

Il-6 is a multifunctional cytokine that promotes the secretion of inflammatory factors, such as IL-1, exerting an inflammatory effect. On the other hand, it could also promote the form of anti-inflammatory factors, such as IL-1ra, exerting an anti-inflammatory effect. Because orthodontic tooth movement is a chronic inflammatory process, Guo et al. speculated that IL-6 mainly exerts an inflammatory effect on orthodontic tooth movement [36]. The current study also investigated the relationship between the SNP for the IL-6 gene and EARR occurrence. The presence of the C/G variant for the IL-6 gene was comparable in those with and without EARR (12.50% and 9.84%, respectively). This result is in contrast with a previous report by Guo et al. [36], where it was found that the IL-6 SNP rs1800796 was one of the more frequently detected SNPs in Han Chinese; moreover, multiple linear regression analysis showed that the IL-6 SNP rs1800796 GC was a risk factor for EARR [36]. Comparison of the CC and GC genotypes for the IL-6 SNP rs1800796 showed that the extent of root resorption in patients with the GC genotype was greater than in the patients with the CC genotype [36]. However, Guo et al. chose the central incisors as their target, although results from the previous studies showed that EARR occurrence is higher in lateral incisors than central incisors or molars [25,26,27,37,38]. This choice, as well as the 3-dimensional measurements method used on the Han population, could have influenced the different measurements obtained in our study based on the Linge Lange 2D method in a Caucasian population. Furthermore, there was also a difference in another characteristic of the group: the age distribution of the 174 subjects was skewed (mainly concentrated at 12–16 years) comparing to the mean age of our population, which was 22 years/9 months (±6 years/3 months).

Since IL-6 may plausibly exert both pro-inflammatory (by promoting the secretion of IL-1) and anti-inflammatory (by promoting the release of IL-1ra) effects, its role on EARR occurrence still needs to be clarified.

### 4.4. Strengths and Limitations

Few studies with an adequate sample size in order to reach at least a statistical power of 80% were conducted in order to investigate the association between SNPs and EARR occurrence. Moreover, the creation of a multi-level model allowed us to quantify the strength and not only the presence/absence of association between genetic variants and root resorption. Nonetheless, some limitations should be highlighted. First of all, EARR occurrence was evaluated in OPG radiographs instead of 3-dimensional cone-beam computer tomography (CBCT) scans, which was regarded as the most accurate tool to evaluate even the slightest changes in root length [26]. Moreover, EARR occurrence was evaluated as the relative change in root-crown ratio and not as the millimeters of root resorption. Consequently, the type of outcome measurement and the tool used to measure it might have undervalued EARR occurrence. Moreover, individuals were enrolled from a private practice instead of a public environment; therefore, the generalizability of the results might have been reduced. Further studies are needed to elucidate the impact of genetic variants related to other molecular pathways involved in EARR development. Furthermore, a sound knowledge of the SNPs behind EARR development may pave the way for the application of gene therapy approaches to root resorption. Indeed, such approaches, which include the potential administration of proteins influencing osteoclasts’ activity, could be implemented in order to reduce the risk of root resorption in those subjects susceptible to developing EARR [39].

## 5. Conclusions

Within the limitations of this study, we demonstrated a significant association between EARR occurrence and the SNP for the IL-1β gene. Conversely, the effect of SNPs for CASP1, TNFRSF11B, and IL-6 genes on EARR presence was not confirmed by the present study. There is currently no accurate method to determine which patients will develop EARR prior to orthodontic treatment. Further studies are needed to investigate the predictive ability of further genetic variants on EARR development.

## Figures and Tables

**Table 1 jcm-10-04166-t001:** Descriptive statistics of patients’ characteristics and genetic polymorphisms.

Variable	Mean ± SD (Proportion)	95% CI	
		Lower	Upper
Age	21.32 ± 7.28	19.88	22.75
Gender			
Males	23.76%	16.51	32.93
Females	76.24%	67.07	83.48
EARR *			
Present	39.60%	30.62	49.35
Absent	60.40%	50.65	69.38
**Single-Nucleotide Polymorphisms**			
IL-1β			
Present (G/A)	8.91%	4.76	16.07
Absent (G/G or A/A)	91.09%	83.93	95.24
TNFRSF11B			
Present (C/T)	23.76%	16.52	32.93
Absent (T/T or C/C)	76.24%	67.07	83.48
CASP1			
Present (C/T)	20.79%	14.02	29.70
Absent (T/T or C/C)	79.21%	70.30	85.98
IL-6			
Present (C/G)	10.89%	6.19	18.46
Absent (G/G)	89.11%	81.54	93.81

Abbreviations: SD, standard deviation; CI, confidence interval; EARR, external apical root resorption; IL-1β, interleukin-1 beta; TNFRSF11B, tumor necrosis factor receptor superfamily, member 11b; CASP1, caspase 1; IL-6, interleukin-6; G/A, guanine-adenine; G/G, guanine-guanine; A/A, adenine-adenine; C/T, cytosine-thymine; T/T, thymine-thymine; C/C, cytosine-cytosine; C/G, cytosine-guanine. * The presence of EARR was judged as: present if rRCR < 0.90, absent if 0.90 ≤ rRCR ≤ 1.00.

**Table 2 jcm-10-04166-t002:** Patient- and tooth-level variables and genetic polymorphisms according to EARR presence.

Variable	EARR		*p*-Value *
	EARR, Yes (rRCR < 0.90)	EARR, No (0.90 ≤ rRCR ≤ 1.00)	
	*n* = 40	*n* = 61	
Age	19.93 ± 6.88	22.23 ± 7.44	0.10
Gender			
Males	15%	29.51%	0.15
Females	85%	70.49%	
Tooth type			
Incisor	88.17%	79.17%	0.04
Molar	11.83%	20.83%	
Arch			
Upper	52.69%	60.74%	0.15
Lower	47.31%	39.26%	
Sites			
Right upper arch			0.06
Central incisor	8.60%	10.14%	
Lateral incisor	20.43%	8.94%	
First molar	1.08%	10.91%	
Left upper arch			
Central incisor	8.60%	10.14%	
Lateral incisor	12.90%	9.71%	
First molar	1.08%	10.91%	
Right lower arch			
Central incisor	10.75%	9.92%	
Lateral incisor	10.75%	9.92%	
Left lower arch			
Central incisor	15.05%	9.49%	
Lateral incisor	10.75	9.92%	
**Single-Nucleotide Polymorphisms**			
IL-1β			
Present	17.50%	3.28%	0.03
Absent	82.50%	96.72%	
TNFRSF11B			
Present	80%	26.23%	0.63
Absent	20%	73.77%	
CASP 1			
Present	22.50%	19.67%	0.80
Absent	77.50%	80.33%	
IL-6			
Present	12.50%	9.84%	0.75
Absent	87.50%	90.16%	

Abbreviations: EARR, external apical root resorption; rRCR, relative changes of root-crown ratio; IL-1β, interleukin-1 beta; TNFRSF11B, tumor necrosis factor receptor superfamily, member 11b; CASP1, caspase 1; IL-6, interleukin-6. * Level of significance set at 5%.

**Table 3 jcm-10-04166-t003:** Intercept-only model to demonstrate the evidence of significant clustering of teeth (Level 1) according to patient-level (Level 2). Moreover, results of the multi-level binary logistic model with EARR as the dependent variable are presented together with the following predictors: tooth type (Level 1), lower arch (Level 1), and IL-1β polymorphism (Level 2).

**Intercept-Only Model (ICC = 0.480)**							
**LR Test vs. Logistic Model**		**chi^2^**	**Prob > chi^2^**				
		71.47	0				
**EARR, Present** **(rRCR < 0.90)**	**Coefficient**	**SE**	**z**		***p*-Value ***	**95% CI**	
						**Lower**	**Upper**
_cons	−3.282	0.309	−10.62		0	−3.889	−2.675
Patient (group variable)							
var(_cons)	3.021	1.001				1.589	5.807
**Final model**							
**Wald chi^2^**		**Prob > chi^2^**					
11.06		0.01					
**EARR, Present** **(rRCR < 0.90)**		**OR**	**SE**	**z**	***p*-Value ***	**95% CI**	
						**Lower**	**Upper**
Tooth type, molar		0.405	0.149	−2.45	0.014	0.196	0.834
Lower arch		1.633	0.410	1.95	0.051	0.998	2.671
IL-1β polymorphism		3.531	2.910	1.53	0.046	1.011	17.765
_cons		0.031	0.011	−9.89	0.000	0.015	0.061
Patient (group variable)							
var(_cons)		3.287	1.020			1.615	5.903

Abbreviations: ICC, intraclass correlation; LR, likelihood ratio; EARR, external apical root resorption; rRCR, relative changes in root-crown ratio; SE, standard error; CI, confidence interval; IL-1β, interleukin-1 beta. * Level of significance set at 5%.

## Data Availability

The data presented in this study are available on request from the corresponding author. The data are not publicly available due to privacy.

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
