# Peer review of "Association between Polymorphisms in the IL-1β, TNFRSF11B, CASP1, and IL-6 Genes and Orthodontic-Induced External Apical Root Resorption"

_jcm, 2021, doi:10.3390/jcm10184166_

Round 1
Reviewer 1 Report
This paper describes that association between polymorhisma in the IL-1B, TNFRF11B, CASP1 and IL-6 genes and orthodontic-induced external root resorption.
Generally, this manuscript is interesting. However, there are some concerns as presented and some of these are discussed below.
Major Comments:
- The association between EARR and the SNPs for the IL-1B gene already reported. Furthermore, that of IL-6 and CASP1 genes also reported by Guo et al. and Sharab et al.. The authors should consider these differences in detail.
- Were the samples in this study extraction case? The amount of tooth movement and torque is different according to the case. The authors should describe about the treatment contents.
- I hope that the authors examine about the length and amount of root resorption by CBCT, and more than 100 cases.
Reviewer 2 Report
- Abstract needs extensive English language editing
- In page 2, starting from the 2nd paragraph, briefly mention the factors affecting EARR and move paragraphs 2-5 to discussion
- In section 2.2, describe the inclusion and exclusion criteria. Do not refer to other articles
- In Examiners calibration section, the mentioned number is not indicative. Calculate intra and inter examiner reliability using intraclass correlation coefficient (ICC).
- The prevalence of EARR in this study seems to be very high compared to the norm. Please elaborate. The use of OPG radiograph may not be accurate
- In page 6, “Detailed data about the extent and severity of EARR in the examined teeth are reported in a previous publication”, please mention the data. Do not refer to other article
- Add more content to discussion regarding IL-1B, which is the only variable with significant effect
Round 2
Reviewer 1 Report
I am now satisfied that all necessary changes have been made.
Further, the comments of other referees were cleared in this revised manuscript.
Author Response
Thank You for Your comments.
Reviewer 2 Report
- In abstract, replace “The aim of the present study was to assess the genetic influence on postorthodontic EARR of selected single nucleotide polymorphisms (SNPs) in four genes: IL1B gene, TNFRSF11B gene, CASP1 gene, IL-6 gene” with “The aim of the present study was to assess the influence of selected single nucleotide polymorphisms (SNPs) IL1B, TNFRSF11B, CASP1, and IL-6 genes on postorthodontic EARR’
- In page 2, merge paragraph 2-4 to one paragraph.
- The age range, 18 and 70 years old, is very broad. Why such wide age range was selected? How old was the oldest patient? This wide age range might affect the results
- In section 3.2, please mention in the text that these data are previously reported somewhere else, not only with mentioning the reference number
- In conclusion, both in abstract and in the text, add “within the limitations of this study”
- In discussion, talk more about possible gene therapy approaches in orthodontics. For example the bellow reference notes that “Administration of proteins that affect or activate osteoclasts could be a direct approach to modulate tooth movement and reduce root resorption”. Please use and cite this reference and other similar research
Atsawasuwan et al. "Advances in Orthodontic Tooth Movement: Gene Therapy and Molecular Biology Aspect." Current Approaches in Orthodontics. IntechOpen, 2018.
